# Estimation of 1-Repetition Maximum Using a Hydraulic Bench Press Machine Based on User’s Lifting Speed and Load Weight

**DOI:** 10.3390/s22020698

**Published:** 2022-01-17

**Authors:** Jinyeol Yoo, Jihun Kim, Byunggon Hwang, Gyuseok Shim, Jaehyo Kim

**Affiliations:** 1Department of Advanced Convergence, BK21 FOUR, Handong Global University, Pohang 37554, Korea; itzy@handong.edu (J.Y.); bts@handong.edu (B.H.); 2Department of Mechanical and Control Engineering, Handong Global University, Pohang 37554, Korea; 21738003@handong.edu (J.K.); 21500380@handong.edu (G.S.)

**Keywords:** 1RM, hydraulic exercise equipment, weight training, health, fitness

## Abstract

1-repetition maximum (1RM), a representative index for an individual’s weightlifting capacity, provides an organized workout guide, but to measure 1RM needs several repetitive exercises up to one’s limit and has a risk of injury, thus, not adequate for beginners, elders, or disabled people. This study suggests a simpler and safer 1RM measurement method using a hydraulic fitness machine. We asked twenty-five female subjects with less than a month of experience in weight training to repeat chest exercises using a conventional plate-loaded bench press machine and a hydraulic bench press machine and measured 1RMs. Repeated-measures ANOVA and paired *t*-test reported the difference between the plate and hydraulic 1RMs insignificant (*p*-value = 0.082) and confirmed the generality of 1RM across the different types of fitness machines. We then derived several 1RM equations in terms of load weight W and lifting speed v during non-1RM exercise and reduced it to a first-order polynomial expression 1RM=−0.3908+0.8251W+0.1054v with adjusted R-square of 0.8849. Goodness-of-fit test and comparison with 1RM equations from reference studies ($v=-1.46\times \frac{W}{1\mathrm{RM}}+1.7035,\;\frac{W}{1\mathrm{RM}}\times 100=7.5786v^{2}-75.865v+113.02$) verified our formula valid. We finally simplified the 1RM measurement process up to a maximum of three repetitions.

## 1. Introduction

With rising health awareness, the global health and fitness club market reached USD 96.7 billion in 2019 with an annular growth rate of 8.7%, and about 184 million people worldwide go to fitness club [1,2]. Fitness clubs usually measure trainees’ 1-repetition maximum (1RM) and design customized weight training routines to increase workout effects according to their athletic abilities [3]. 1RM is the maximum weight that a person can lift once and is often used as a representative index to evaluate an individual’s weightlifting capacity [4]. 1RM is usually measured by exercising once with the smallest load weight and increasing it up to one’s limit [5]. Studies report that the average bench press 1RMs of male and female adults with more than 3 months of exercise experience are 98 kg and 39 kg, and 2–3 sets of 12 or more repetitions with a load weight no greater than 67% of 1RM are recommended for enhancing muscular endurance [6,7]. However, 1RM measurement is not an easy task, especially for workout beginners, elders, disabled people, or patients undergoing rehabilitation. Since a user repeats exercises up to one’s limit, there is a high risk of injury; 2.4% to 7.6% of weight training population get injured every year [8]. In particular, the number of elderly men injured during weight training has tripled over the past 20 years [9].

Recent studies suggest relatively safer methods to measure 1RM using mathematical equations defined in terms of load weight and lifting speed during a normal exercise [10,11]. They report 7% and 9.3% average 1RM errors, so a user can get fairly accurate 1RM without repeating the exercise to the limit. On the other hand, workout machines using hydraulic cylinders, such as rowing simulators or steppers, have recently been used in sports and rehabilitation centers [12]. Unlike conventional machines that use weight plates and work against gravity, hydraulic fitness machines can easily control their load weights by simply turning valve knobs. Since there is no physical weight involved with the workout other than viscous damping which resists rate of change in motion, users of hydraulic fitness machines are less likely to get injured during workout. Additionally, since they offer similar exercise effects to conventional machines, more hydraulic exercise equipment is widely adopted [13].

In our previous research, we proposed an intelligent hydraulic bench press machine [14]. The system detected a user’s tiredness and automatically adjusted the load weight, so it played a similar role as a workout trainer. In this study, we measured 1RMs from twenty-five subjects using both conventional press machine and our hydraulic press machine and observed statistical differences between the paired 1RM. Then, we derived several 1RM polynomial equations in terms of load weight and lifting speed for the hydraulic press machine. After testing goodness-of-fit of the models, we selected one model as a valid 1RM equation and suggested a concise 1RM measurement process.

## 2. Materials and Methods

### 2.1. Research Method

When exercising with a conventional bench press machine, a user lies down on a flat bench and lifts a handle. A cable connects the handle with a set of 4.54 kg weight plates. As shown in Figure 1a, we modified the device to a hydraulic press machine by attaching a hydraulic cylinder (U2W-Type A, WATA Corporation, Paju-si, Korea) to the handle, so viscous damping acts as a lifting load instead of the weight plates. Then, a step motor (iHSS57-36-10, Shenzhen Just Motion Control Electromechanics Co., Shenzhen, China) was coupled to the valve knob. With a computer program, we could precisely rotate the knob and control the hydraulic pressure to a desired magnitude in real-time. We attached a variable resistor to the handle pivot to calculate the average speed taken during each lift.

Figure 1b depicts an input-output characteristic of the hydraulic cylinder. First, the load weight caused by hydraulic pressure increases from 10 to 115 kg as the dial indicator of the cylinder changes from 1 to 8, so it has a suitable range for a beginner’s bench press exercise [15]. Since the hydraulic cylinder acts as a damper in the system, the load weight also increases with the increase in extension speeds of the cylinder rod. With the current angular position of the step motor and voltage across the variable resistor, we computed the current dial indicator $y_{n}$ and average extension speed v during each lift. Thus, we calculate the average load weight W felt by the subject using the following input-output equations derived from Figure 1b:(1)$$W=\{\begin{array}{l} 6.25\;y_{n}+3.75,\\ 11.79\;y_{n}-10.66,\\ 12.64\;y_{n}+13.35, \end{array}\;\;\;\begin{array}{l} v=50\;\mathrm{mm}/s,\;\;\;y_{n}<2.6\\ v=50\;\mathrm{mm}/s,\;\;\;y_{n}\geq 2.6\\ v=100\;\mathrm{mm}/s \end{array}$$

### 2.2. Experiment Methods

#### 2.2.1. Subject

Twenty-five female subjects (20.97±1.43 years) with less than a month of weight training experience participated in our study. We asked the subjects to do bench press exercise once in every week for a month. In advance, we informed the purpose and plan of the experiment and obtained consent.

#### 2.2.2. Experimental Design

In the first week, we measured each subject’s 1RM using a conventional bench press machine. We asked the subjects to do bench press exercise one time with the lightest load weight of 9.07 kg. If successful, we added a 4.54 kg weight plate to the machine and asked to repeat the exercise. If unsuccessful, we stopped the experiment and recorded the final successful load as 1RM. We gave a two-minute break between each trial and waited for the subjects to recover their maximum isometric force [16].

For the following three weeks, we measured each subject’s 1RM three times using the hydraulic bench press machine. Instead of increasing the load weight by adding 4.54 kg plates to the press machine, we designed a similar 1RM measurement process for the hydraulic press machine summarized in Figure 1c. We started the experiment by requesting the subjects to do press exercise one time with a dial indicator of 1. If successful, we increased the weight load by turning the valve knob and asked to repeat the exercise. We rearranged the previous input-output equations and determined the next dial indicator $y_{n+1}$ based on the current dial indicator $y_{n}$ and the user’s average lifting speed v:(2)$$y_{n+1}=\{\begin{array}{c} 0.0205y_{n}v+0.0307v-0.0230y_{n}-1.5363,\;\;\\ 0.0014y_{n}v+0.0407v+0.9278y_{n}-2.0362,\;\; \end{array}\;\;\begin{array}{l} y_{n}<2.6\\ y_{n}\geq 2.6 \end{array}$$

If the load difference between the next and current dial indicators was less than 4.54 kg, we concluded that the change in the load weight is negligible. Thus, we stopped the press exercise according to the following termination condition and recorded the final successful load as 1RM:(3)$$\{\begin{array}{l} y_{n+1}-y_{n}<0.70,\\ y_{n+1}-0.53y_{n}<1.60,\\ y_{n+1}-y_{n}<0.37, \end{array}\;\;\;\begin{array}{l} y_{n+1}<2.6,\;\;\;y_{n}<2.6\\ y_{n+1}\geq 2.6,\;\;\;y_{n}<2.6\\ y_{n+1}\geq 2.6,\;\;\;y_{n}\geq 2.6 \end{array}$$

The entire data were collected using an analog-digital convertor (National Instrument USB-6009) and processed using MATLAB R2019a (The MathWorks, Inc.). We used IBM SPSS Statistics 21 (IBM Corp.) to perform a repeated-measures ANOVA and a paired *t*-test between the two direct 1RMs from the conventional and hydraulic bench press machines. Since the data did not have heterogeneity of variances, we used Greenhouse–Geisser correction. We used R to suggest three indirect 1RM equations and to statistically validate their coefficients [17].

## 3. Results

### 3.1. Comparison between Plate and Hydraulic 1RMs

Figure 2 illustrates a representative subject’s raw data during press exercises using the hydraulic press machine with three different dial indicators. With a dial indicator of one, the lowest reading, the subject successfully lifted the handle in about one second, shown in Figure 2a. Since the lift was fast enough, we could significantly increase the dial indicator to 3.11 and suggest a heavier load according to the algorithm in Figure 1c. As the press load increased, the lifting speed gradually decreased as in Figure 2c. The mean velocity of 40.74 mm/s computed no meaningful change in the dial indicator, therefore, we could decide 1RM of the subject as 48.94 kg.

Figure 3 summarizes all the 1RMs obtained using the conventional and hydraulic press machines. We conducted a repeated-measures ANOVA on the data to examine the paired differences across the plate 1RM and the hydraulic 1RMs. The *p*-value of 0.009 in Table 1 implies that at least one group is significantly different from others. The pairwise comparisons test in Table 2 reports that the plate 1RM and the third hydraulic 1RM are statistically different with the *p*-value of 0.037. However, we cannot easily reject the null hypothesis because the *p*-value is close to the significance level of 0.05 [18]. Since a few additional data would flip the test result, we should withhold the decision on this case. Even if we accept the alternative hypothesis, the two other hydraulic 1RMs show no differences from the plate 1RM, in contrast. Therefore, we have a disagreement among the repeated-measures ANOVA. By closely observing the mean differences in Table 2, we find that the first hydraulic 1RM is significantly smaller than the second and third hydraulic 1RMs. In addition, the average hydraulic 1RM tends to increase over the time. One possible explanation to the disagreement is that the subjects became accustomed to the hydraulic press machine and to chest exercise.

To neglect data inconsistency, we calculated mean hydraulic 1RM and conducted a paired *t*-test with the plate 1RM. The test result in Table 3 reveals the *p*-value of 0.082, implying there is no statistical difference. Thus, we carefully conclude that 1RMs measured using the hydraulic bench press machine are not different from 1RMs measured using the conventional machine.

### 3.2. RM Equation

We separated twenty-five non-1RM data sets and derived three different polynomial equations in terms of load weight W and lifting speed v as shown in Table 4.

Although Model 1 shows small residual standard error and high adjusted R-square, the coefficient of second-order term of the load weight is insignificant. We discarded the term and derived Model 2. All parameters in Model 2 became statistically significant, and the magnitudes of residual standard error and adjusted R-square did not change. We performed model reduction one more time by removing the second-order term of the lifting speed to have a first-order linear 1RM equation. Model 3 shows statistical significance in both load weight and lifting speed. Although Model 3 has a simpler expression than Model 2, it has 12.6% more residual standard error and 2.9% lower adjusted R-square.

We tested the goodness-of-fit on the two models and they are presented in Figure 4. From the first graphs, we observe that Model 3 has more homogeneity of variance because data points are more randomly scattered around the zero residual. From the second graphs, we see that Model 3 has a stronger normality since the data points fit in a line. We observe a bias in Model 2 at the positive extreme. No major difference between the models was observed from the third and the fourth graphs. Thus, we concluded that Model 3 is more valid to estimate 1RM and indicated the graph of Model 3 through Figure 5.

During the 1RM measurement, the subject had to repeat the lifting exercise up to 11 times using the conventional press machine and 7 times for the hydraulic. Since we have derived a valid 1RM equation, we can considerably reduce the repetitions up to 2 or 3 times to estimate 1RM by suggesting an appropriate starting load weight. We rearranged the hydraulic 1RMs in Figure 3 with the subjects’ lifting speeds in Table 5. We followed Scott’s rule to decide the ranges of 1RM and lifting speed [19].

Considering that 40 to 100 mm/s is an appropriate lifting speed during chess press exercises, suggesting 25 kg as a starting load weight for inexperienced female users is reasonable because a total of 7 + 10 + 9 = 26 out of 75 or 34.67% of them would get their 1RM calculated at once. The other 26 out of 75 might feel the load weight heavy so the lifting speeds would be under 40 mm/s, whereas the remaining 23 out of 75 might feel light producing the lifting speeds over 100 mm/s. For the second trial, we could adjust the load weight to 15 or 35 kg according to the previous lifting speeds. Then, about 45.33% of users would get their 1RM measured by the equation. Likewise, the users at the extremes would have their 1RMs after the third trial. Figure 6 illustrates the simplified 1RM measurement algorithm using the hydraulic press machine.

## 4. Discussion

### 4.1. Comparison between Plate and Hydraulic 1RM Equations

In Results, we have derived the following 1RM equation for the hydraulic press machine in terms of load weight W and lifting speed v:(4)1RM=−0.3908+0.8251W+0.1054v

Two 1RM equations which use the same independent variables but derived from conventional press machines were selected from reference papers [10,11]:(5)$$v=-1.46\times \frac{W}{1\mathrm{RM}}+1.7035$$
(6)$$\frac{W}{1\mathrm{RM}}\times 100=7.5786v^{2}-75.865v+113.02$$

We depicted the two reference equations and their decision boundaries in Figure 7. A notable difference in Figure 5 and Figure 7 is that the slope of our 1RM equation is tilted towards left, so it reflects the lifting speeds to the estimated 1RM more greatly than other formulas. Thus, given the same load weight, our equation calculates heavier 1RM if the lifting velocity is faster. In addition, we observed that the reference equations have slight offset errors. The decision boundaries in Figure 7 should be increase by 5 to 10 kg to match the colors of the data points and boundaries. Table 6, which summarizes descriptive statistics of 1RM errors calculated from the equations, also shows that the average 1RM errors of the reference equations are 5.024 and 4.208, respectively. From the observations, we can infer that hydraulic 1RMs are generally greater than plate 1RMs by 5 kg on average, but the difference is statistically insignificant as we have already examined from the repeated-measures ANOVA and the paired *t*-test in Table 2 and Table 3. Thus, the negative slope of the decision boundary in Figure 5 is reasonable because it compensates the relatively higher magnitude of hydraulic 1RMs.

### 4.2. Advantages of Using Hydraulic Fitness Machines

In addition to their smaller sizes than conventional fitness machines, hydraulic fitness machines have several notable advantages as follows.

Since the hydraulic cylinder is a damper, the load weight varies proportional to ex-tension speed, as seen in Figure 1b. Thus, the press loads of the hydraulic machine also vary according to users’ lifting speeds. This odd characteristic is actually advantageous. Most conventional fitness machines using weight plates involve a series of lifting and lowering the physical mass against gravity. Many injuries occur when a user gets tired and loses strength in the middle of a workout because there still is a mass to surrender carefully. On the other hand, when using hydraulic fitness machines, a tired user can either slow down to purposely decrease the load weight and continue a workout or simply release the handle to stop exercise because there is no actual weight. Therefore, it provides a safer at-home weight training environment for beginners.

Using hydraulic fitness machines and the indirect measurement method, a user may easily check one’s 1RM with workout data. Based on the 1RM, the machine could suggest an everyday exercise routine according to the user’s exercise purposes, as shown in Table 7.

The research about muscle strength and muscle contraction reports that the relationship between muscle strength and muscle contraction is inversely proportional [20]. In other words, producing high force slows muscle contraction, and shortened muscle contraction limits muscle strength production. In addition, if a load is too light, the muscle contraction speed increases, and the faster the muscle contraction, the smaller the force that the muscle can apply. Through these results, it is necessary to exercise at an appropriate velocity to efficiently produce muscle strength. The other research about muscle force and muscle velocity reports that a person produces maximum power at a value of 40–80% of the maximum velocity [21]. Considering Table 7 and the algorithm proposed in Section 3.2, this can be used to present the ideal indicator according to the target velocity, and it is expected to present a muscle strengthening exercise routine for beginners, elders, disabled people, or patients undergoing rehabilitation. Thereafter, it is possible to easily obtain the 1RM of user and perform at-home weight training accordingly by using hydraulic cylinders not only for bench press machines but also for other exercise equipment.

Additionally, we plan to improve the hydraulic fitness machine to expand the target users to paralyzed patients and disabled people. Thus, as we have estimated 1RMs, we expect to evaluate their physical abilities quantitatively and help rehabilitation exercises.

## Figures and Tables

**Figure 1 sensors-22-00698-f001:**
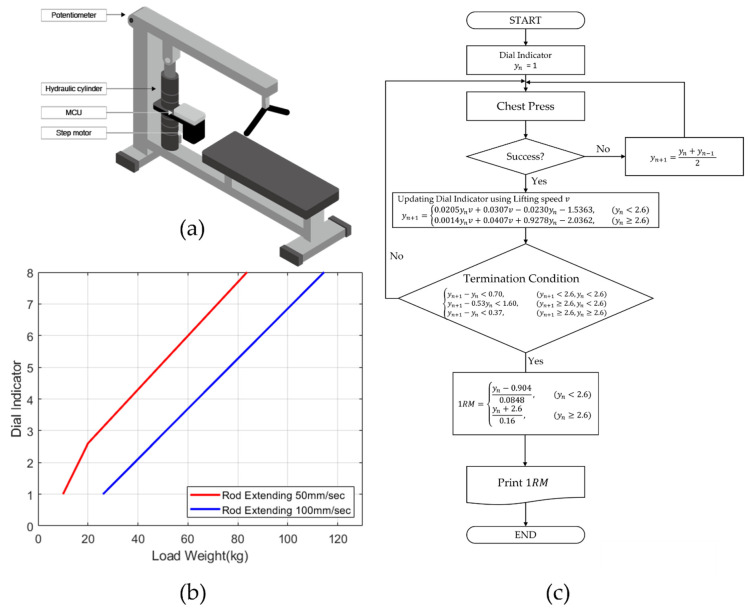
(**a**) A hydraulic cylinder and a step motor were connected to a conventional bench press machine. Instead of weight plates, hydraulic pressure from the cylinder acted as load weight to the machine; (**b**) As the dial indicator increases, the load weight almost linearly increases. The extending speed of the cylinder rod affects the load weight as well; (**c**) The flow chart summarizes the 1RM measurement process using the hydraulic press machine.

**Figure 2 sensors-22-00698-f002:**
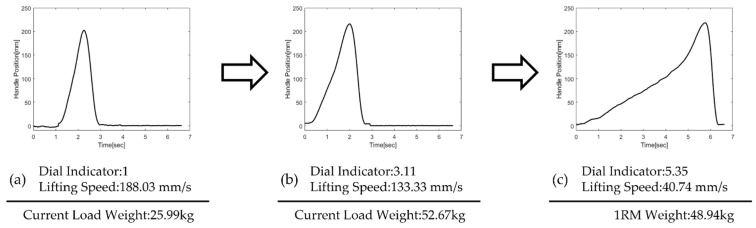
The handle position of the bench press machine during lifting exercises with the dial indicator of (**a**) 1, (**b**) 3.11, (**c**) 5.35. If the chest press exercise is successful and fast, the dial indicator of hydraulic cylinder valve increases and suggests heavier load weight until the lifting speed naturally decreases and the load weight reaches 1RM.

**Figure 3 sensors-22-00698-f003:**
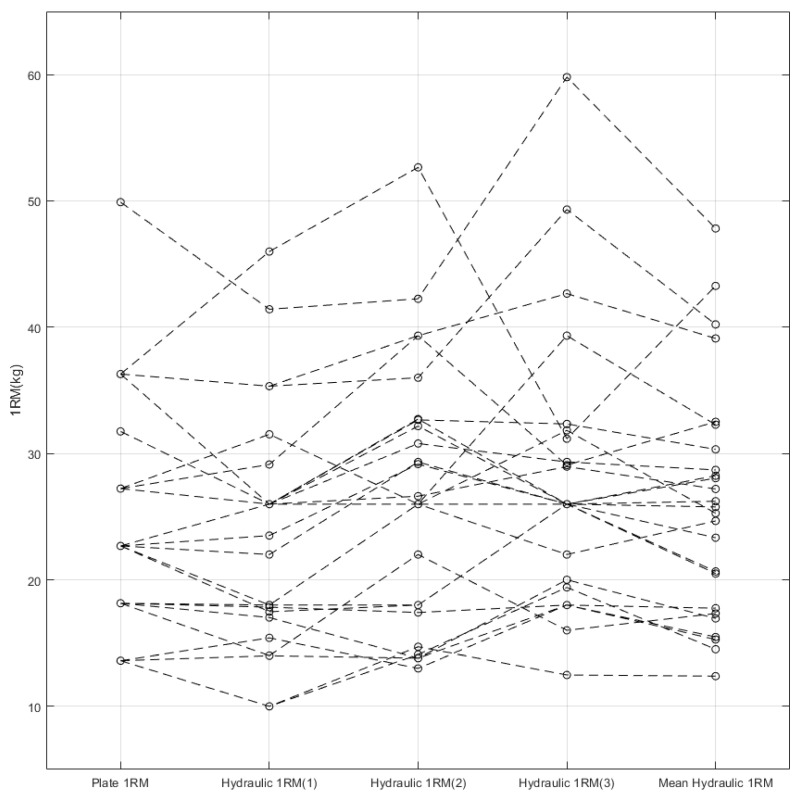
1RM measured using conventional press machine (plate 1RM) and 1RM measured using hydraulic press machine (hydraulic 1RM). Mean hydraulic 1RM represents the average of the three paired hydraulic 1RMs.

**Figure 4 sensors-22-00698-f004:**
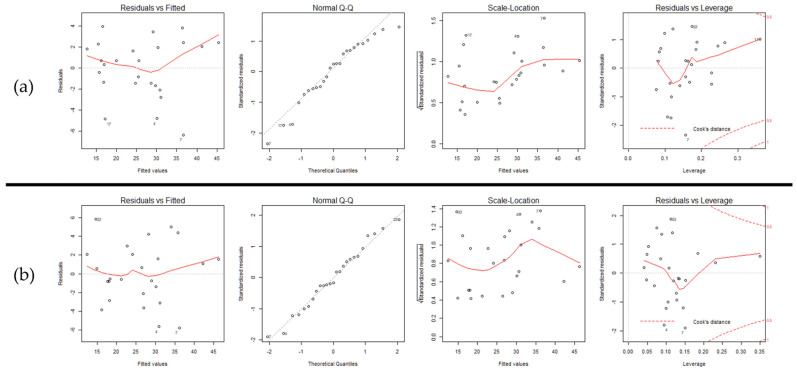
Goodness-of-fit of the two 1RM equations’ load weight W and lifting speed v: (**a**) $1\mathrm{RM}=-14.8408+0.8644W+0.6040v-0.0039v^{2}$; (**b**) 1RM=−0.3908+0.8251W+0.1054v.

**Figure 5 sensors-22-00698-f005:**
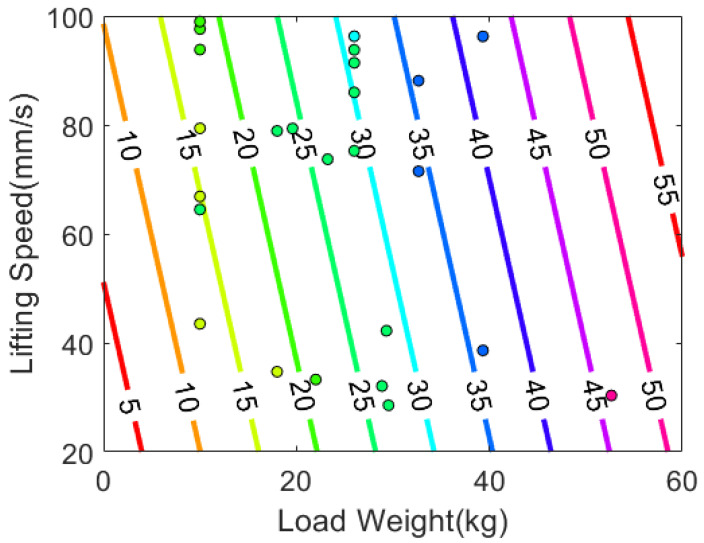
Non-1RM data points and 1RM decision boundaries based on the equation, 1RM=−0.3908+0.8251W+0.1054v. The color of each data point indicates the actual 1RM.

**Figure 6 sensors-22-00698-f006:**
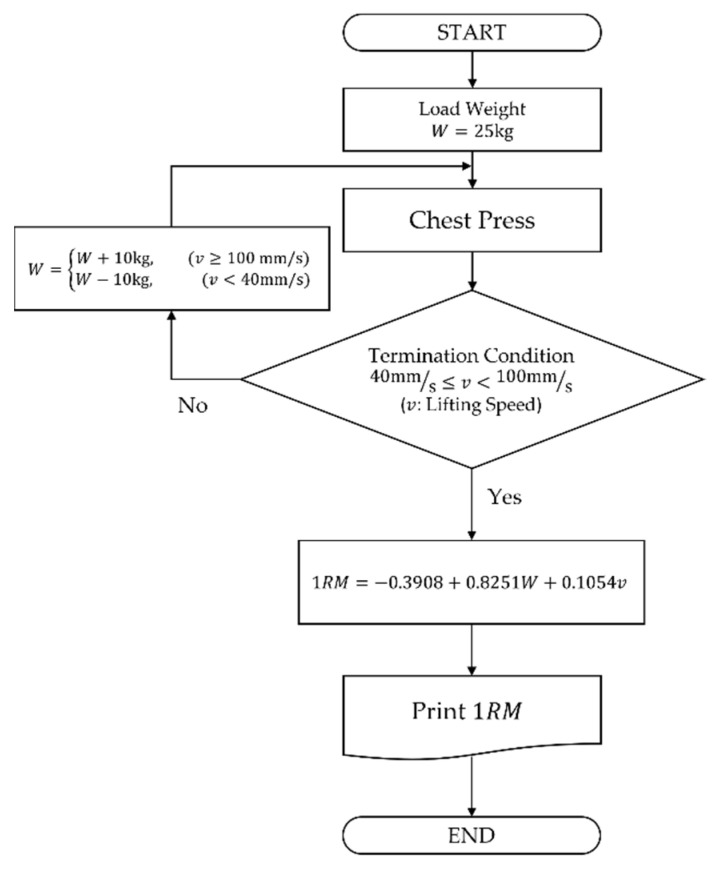
Simplified 1RM measurement algorithm using the hydraulic press machine.

**Figure 7 sensors-22-00698-f007:**
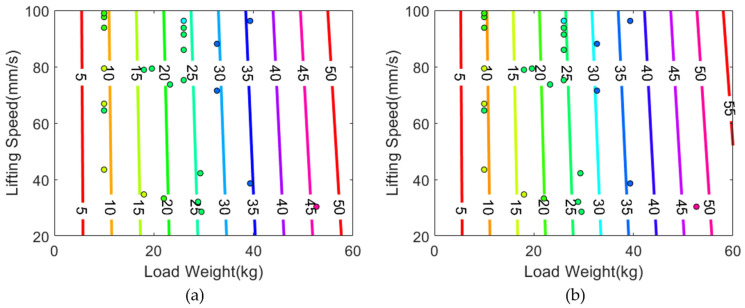
Non-1RM data points and 1RM decision boundaries based on the equations from reference papers: (**a**) $v=-1.46\times \frac{W}{1\mathrm{RM}}+1.7035$; (**b**) $\frac{W}{1\mathrm{RM}}\times 100=7.5786v^{2}-75.865v+113.02$. The color of each data point indicates the actual 1RM.

**Table 1 sensors-22-00698-t001:** Repeated-measures ANOVA of plate 1RM and hydraulic 1RMs.

Measure	Mean Square	F	Significance ^b^
1RM	165.589	5.709	0.009

^b^ Greenhouse–Geisser correction was used because of non-sphericity.

**Table 2 sensors-22-00698-t002:** Paired comparison of plate 1RM and hydraulic 1RMs.

Measure 1	Measure 2	Mean Difference	Standard Error	Significance ^b^	95% Confidence	
					Lower	Upper
plate 1RM	hydraulic 1RM (1)	1.445	0.829	0.566	−0.939	3.829
	hydraulic 1RM (2)	−1.832	1.086	0.628	−4.956	1.291
	hydraulic 1RM (3)	−2.946	0.982	* 0.037	−5.769	−0.123
hydraulic 1RM (1)	hydraulic 1RM (2)	−3.277	0.829	* 0.004	−5.660	−0.894
	hydraulic 1RM (3)	−4.390	1.295	* 0.014	−8.113	−0.667
hydraulic 1RM (2)	hydraulic 1RM (3)	−1.113	1.659	1.000	−5.883	3.656

Based on the estimated marginal means. * The mean difference is significant at the 0.05 level. ^b^ Adjustment for multiple comparisons: Bonferroni.

**Table 3 sensors-22-00698-t003:** Paired *t*-test of plate 1RM and mean hydraulic 1RM.

	Mean	Standard Deviation	Standard Error Mean	95% Confidence		*t*	df	Significance (2-Tailed)
				Lower	Upper			
plate 1RM—mean hydraulic 1RM	−1.11120	3.06384	0.585	−2.37589	0.15349	−1.813	24	0.082

**Table 4 sensors-22-00698-t004:** 1RM equations and model reduction.

Model	Parameter	Order	Coefficient	*p*-Value	Residual Standard Error	Adjusted R-Squared
1	Intercept		−13.965266	* 0.03552	2.969	0.8995
	Load weight, W (kg)	1	0.703188	** 0.00261		
		2	0.003117	0.42045		
	Lifting speed, v (mm/s)	1	0.630198	** 0.00427		
		2	−0.004075	* 0.01379		
2	Intercept		−14.840838	* 0.02308	2.946	0.901
	Load weight, W (kg)	1	0.864368	*** 0.00000		
	Lifting speed, v (mm/s)	1	0.603973	** 0.00478		
		2	−0.003890	* 0.01578		
3	Intercept		−0.39076	0.89201	3.317	0.8745
	Load weight, W (kg)	1	0.82505	*** 0.00000		
	Lifting speed, v (mm/s)	1	0.10540	** 0.00113		

Based on the estimated marginal means. * The mean difference is significant at the 0.05 level. ** The mean difference is significant at the 0.01 level. *** The mean difference is significant at the 0.001 level.

**Table 5 sensors-22-00698-t005:** Frequency table of 1RM and corresponding lifting speed using a hydraulic press machine.

Count		1RM (kg)					Total
		(10, 20)	(20, 30)	(30, 40)	(40, 50)	(50, 60)	
Lifting speed (mm/s)	(100, 120)	4	3	1	0	1	9
	(80, 100)	6	7	1	1	0	15
	(60, 80)	11	10	9	3	1	34
	(40, 60)	5	9	2	1	0	17
Total		26	29	13	5	2	75

**Table 6 sensors-22-00698-t006:** Descriptive statistics of 1RM errors calculated from the three 1RM equations.

1RM Error	N	Mean	Standard Deviation	Standard Error	95% Confidence		Minimum	Maximum
					Lower	Upper		
Equation (4)	25	−0.572	2.726	0.545	−1.697	0.553	−4.95	6.42
Equation (5)	25	5.024	2.392	0.478	4.037	6.012	1.62	10.75
Equation (6)	25	4.208	2.299	0.460	3.259	5.157	0.52	9.32

**Table 7 sensors-22-00698-t007:** The load weight, the number of reps for a set, the number of sets and break time depending on the exercising type [7].

Exercising Type	Load Weight	Reps for a Set	The Number of Sets	Break Time per Set
Muscular exercise	Above 85% of 1RM	Under 6 reps	2–6 sets	2–5 min
Muscular hypertrophy exercise	About 67–85% of 1RM	6–12 reps	3–6 sets	30–90 s
Muscle endurance exercise	Below 67% of 1RM	Over 12 reps	2–3 sets	30 s

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
