# Peer review of "Estimation of 1-Repetition Maximum Using a Hydraulic Bench Press Machine Based on User’s Lifting Speed and Load Weight"

_sensors, 2022, doi:10.3390/s22020698_

Round 1

Reviewer 1 Report

I want to congratulate the authors for their effort with the methodology employed and the writing of this paper. I would also like to list some comments so the manuscript can be improved:

  • Line 20 in the abstract: where it says "significantly accurate" it might say "significantly more accurate" or similar as described later on the results.
  • Lines 44 and 45 containing reference [6] are not relevant for the purpose of the study.
  • Figure 1c should be improved to make it more self-explicative or at least extend the information on the footnote so it can be more easily understood.
  • Line 92: "little no experience" might be clarified to "little" or "no" experience.
  • Figure 3: paired t-test have been carried out to find out potential differences between plates and hydraulic bench press. However, this test has considered a single plate measure with an average of three measures carried out with hydraulic material. When having a look on individual's results on the three trials, differences up to >20kg between weeks are found for some of them whose plate-1RM is 35 to 50 kg. Authors must clarify why average hydraulic results should be considered and provide ANOVA tests comparing plate-RM with each single hydraulic trial. If differences where found, authors should explain possible reasons and at least include three hydraulic trials in the full method.
  • General: authors present their results as a useful tool for home workouts, specially in the first lines of the introduction. However, hydraulic machines are not commonly available for most practitioners. Authors might reconsider the information provided in their introduction and the applicability of the equations proposed. These seem to have a potential role in clinical trials, but not in home workouts.
  • Information provided in table 7 is not relevant for the objective of the study. 

Reviewer 2 Report

Generic comments
"Direct and indirect measurement of weight training beginners’ 1-repetition maximum using a hydraulic bench press machine" is an attractive concept. The practical implications of this work may lead to minimizing the likelihood of injury in beginners who are just starting their training. This is a good study and would be of interest to other researchers. The paper is write clearly with a simple and adequate methods. It is well written and organized. 

Specific comments
The aims of the study should be clarified and clearly stated at the end of Introduction and in the further parts of the work in which the authors thoroughly explained the mathematical modeling carried out in the work. The methodology used in the work seems to be correct. The conducted statistical analyzes do not raise any objections, the authors verified all assumptions of the applicability of the analyzes on the basis of which they drew conclusions. Summing up, a very interesting, well-written article, therefore I recommend this article for publication.

Author Response

Thank you for your review

Reviewer 3 Report

Dear Authors,

Thank you for the opportunity to review this article. I understand that the study has taken a lot of time and work but I do not think can be published like it is now.

Major issues.

What is the purpose of the study? To measure directly the 1RM maximum, to measure indirectly? To offer equations for beginners? To improve the hydraulic fitness? To offer a safe weight training at home? To prevent injuries? There are a lot of information but it is not clear the goal of the study.

Title must clearly define the content of the study, the title: “Direct and indirect measurement of weight training beginners’ 1-repetition maximum using a hydraulic bench press machine” does the study explain to the beginners how to do the direct/indirect measurement of 1RM? No, so the title is not correct.

Abstract

It is not clear and uncompleted, where are the equations? Where is the information for the beginners to have a safe weight training at home?

What is this: “We plan to improve the hydraulic fitness machine to expand the target users to paralyzed patients and disabled people to evaluate their physical abilities quantitatively and help rehabilitation exercises” it is not a result from this study.

Which is the conclusion of the study? There is no conclusion

It is ambiguous:

  • what mean: “with little experience in weight training”? it is 1,2,3,4 years? Months?
  • “Equations from reference studies” where are the equations?

Introduction.

Must be improved, sentences like: “Therefore, a workout guide must be suggested to 39 beginners for both safe and organized at-home weight training.” are not explained in the article.

Materials and Methods

In order to calculate the 1RM the subject need to be completely recovered, and it cannot be possible with only “two-minute break between each trial”.

Discussion

It is poor. Where is the discussion? The goal of the discussion is not to present study data; it is just to justified or try to explain (to discuss) the differences between studies. This section should be improved.

 Minor issues

Keywords, do not use words that are already in title, for example “indirect measurement”

The abbreviation should go after the meaning of the abbreviation, 1-repetition maximum (1RM) then why 1RM (Repetition Maximum)?

Why do not use only international system units? 4.4kg(10lbs)

To use only one decimal, for example 21.0 instead of 20.967, there are many unnecessary decimals in the main documents

To add the meaning of all abbreviations in the tables and figures

Incongruity. “little no experiences in weight” (Material and method) or “little experience in weight” (abstract)?

Why the mean of hydraulic 1RM is compered to plate 1RM? Why not 1 hydraulic 1RM? Why no 3 and no 2,4,5…?

Table 6 is confusing, if there are 4 different speed how some of them can be in 2 different speed? For example, 60mm/s below to 40-60mm/s or 60-80mm/s?

Best Regards

Round 2

Reviewer 3 Report

Dear Authors, 

Thank you for your reply. I think the article is better now and can be published. 

Best Regards